# Unsupervised Video Object Segmentation for Deep Reinforcement Learning

**Vik Goel, Jameson Weng, Pascal Poupart**
Cheriton School of Computer Science, Waterloo AI Institute, University of Waterloo, Canada
Vector Institute, Toronto, Canada
{v5goel,jj2weng,ppoupart}@uwaterloo.ca

## Abstract

We present a new technique for deep reinforcement learning that automatically detects moving objects and uses the relevant information for action selection. The detection of moving objects is done in an unsupervised way by exploiting structure from motion. Instead of directly learning a policy from raw images, the agent first learns to detect and segment moving objects by exploiting flow information in video sequences. The learned representation is then used to focus the policy of the agent on the moving objects. Over time, the agent identifies which objects are critical for decision making and gradually builds a policy based on relevant moving objects. This approach, which we call Motion-Oriented REinforcement Learning (MOREL), is demonstrated on a suite of Atari games where the ability to detect moving objects reduces the amount of interaction needed with the environment to obtain a good policy. Furthermore, the resulting policy is more interpretable than policies that directly map images to actions or values with a black box neural network. We can gain insight into the policy by inspecting the segmentation and motion of each object detected by the agent. This allows practitioners to confirm whether a policy is making decisions based on sensible information. Our code is available at *https://github.com/vik-goel/MOREL*.

## 1   Introduction

Tremendous progress has been made in the development of reinforcement learning (RL) algorithms for tasks where the input consists of images. For instance, model-free RL techniques [33] now outperform humans on the majority of the Atari games in the arcade learning environment [3]. Despite this outstanding performance, RL techniques require a lot more interactions with the environment than human players to perform well on Atari games. This is largely due to the fact that RL techniques do not have any prior knowledge of these games and therefore require a lot more interactions to learn to extract relevant information from images. In contrast, humans immediately recognize salient objects (e.g., spaceships, monsters, balls) and exploit texture cues to guess a strategy [6]. Within a few interactions with the environment, humans quickly learn the effects of their actions and can easily predict future frames. At that point, humans can focus on learning a good policy.

Existing RL techniques for image inputs typically use convolutional neural networks as part of a policy network or Q-network. The beauty of deep RL is that there is no need for practitioners to handcraft features such as object detectors, but this obviously requires more interactions with the environment since the convolutional neural network needs to learn what features to extract for a good policy and/or value function.

We propose a new approach that leverages existing work in computer vision to estimate the segmentation and motion of objects in video sequences [8]. In many games, the position and velocity of moving objects constitute important features that should be taken into account by an optimal policy. More

precisely, we describe how to obtain an object mask with flow information for each moving object in an unsupervised fashion based on a modified version of SfM-Net [35] that is trained on the first 1% of the frames seen by the agent while playing. The encoding part of the resulting network is then used to initialize the image encoder of actor-critic algorithms. This has two benefits: 1) the number of interactions with the environment needed to find a good policy can be reduced and 2) the object masks can be inspected to interpret and verify the features extracted by the policy. Since moving objects are not the only salient features in games (e.g., fixed objects such as treasures, keys and bombs are also important), we combine the encoder for moving objects with a standard convolutional neural network that can learn to extract complementary features. We showcase the performance of MOREL on all 59 Atari games where we observe a notable improvement in comparison to A2C and PPO for 26 and 25 games respectively, and a worse performance for 3 and 9 games respectively.

The paper is organized as follows. Sec. 2 provides some background about reinforcement learning. Sec. 3 discusses related work in reducing the number of interactions with the environment and improving the interpretability of RL techniques. Sec. 4 describes our new approach, Motion-Oriented REinforcement Learning (MOREL), that learns to segment moving objects and infer their motion in an unsupervised way as part of a model free RL technique. Sec. 5 evaluates the approach empirically on 59 Atari games. Finally, Sec. 6 concludes the paper and discusses possible future extensions.

## 2 Background

### 2.1 Reinforcement Learning

Reinforcement learning (RL) [33] provides a principled framework to optimize sequences of interdependent decisions in stochastic environments with unknown dynamics. More specifically, an agent learns to optimize a policy $\pi$ that maps histories of observations $h_t = (a_0, o_1, a_1, o_2..., a_{t-2}, o_{t-1}, a_{t-1}, o_t)$ to the next action $a_t$. The value $V^\pi(h) = E_\pi[\sum_{t=0}^{T} \gamma^t r_t]$ of a policy $\pi$ is measured by the expected sum of discounted rewards $r_t$ (with discount factor $\gamma \in [0, 1)$) earned while executing the policy after observing history $h$. Similarly, the action value function $Q^\pi(h, a) = E[r_0|h, a] + E_\pi[\sum_{t=1}^{T} \gamma^t r_t]$ quantifies the expected value of executing $a$ followed by $\pi$. The goal is to find an optimal policy $\pi^* = argmax_\pi V^\pi(h) \; \forall h$ with the highest value function.

RL algorithms can be categorized into value-based techniques (that maximize some type of value function), policy optimization techniques (that directly optimize a policy) and actor-critics (that use both a value function and a policy). In RL problems with complex observation spaces, it is common to use deep neural networks to represent value functions and policies. For instance, in the arcade learning environment [3], observations consist of images and convolutional neural networks [20] are often used to automatically extract relevant features for the selection of actions (policy) and the computation of values (value function) [20, 24, 25, 30, 31, 34].

Policy optimization algorithms include policy gradient techniques [37] that gradually improve a policy by taking steps in the direction of the gradient $\nabla_\theta \pi_\theta(a_t|h_t)R_t$ where $\theta$ consists of the parameters of the policy $\pi$ and $R_t$ is an estimator of the cumulative rewards, $R_t = \sum_k \gamma^k r_{t+k}$. To reduce variance in the estimator, it is common to replace $R_t$ by an estimate $A_t$ of the advantage function, $A(h_t, a_t) = Q(h_t, a_t) - V(h_t)$ [9]. The asynchronous advantage actor critic (A3C) [24] and its synchronous variant (A2C) are good examples of such techniques.

In practice, the policy gradient may not give the best direction for improvement and large steps may induce instability. To mitigate those problems, trust region techniques [30] search for the best improvement to a policy within a small neighbourhood by solving a constrained optimization problem. Within the same spirit, proximal policy optimization (PPO) techniques [31] clip the policy gradient to prevent overly large changes to the policy.

In this paper, we describe an object-sensitive representation learning technique that can be combined with most value-based, policy optimization, and actor-critic algorithms. We will demonstrate the gains in sample efficiency of our object sensitive technique using the Atari domain in the context of A2C and PPO, which are strong baselines that achieve very good results in many domains.

## 2.2 Unsupervised Object Segmentation

In computer vision, it is possible to exploit information induced from the movement of rigid objects to learn in a completely unsupervised way to segment them, to infer their motion and depth, and to infer the motion of the camera. The Structure from Motion Network (SfM-Net) [35] is a good example of a deep neural network that takes as input two consecutive images and computes the geometry of moving objects. This architecture consists of two subnetworks: one which predicts the depth of each pixel in an image, and another which predicts camera motion and masks along with rotation and translation values for moving objects in the image. The combined information is used to compute the optical flow which attempts to reconstruct one of the input frames from the other. By minimizing the $L1$ difference between the reconstructed frame and the true frame, the entire network can be trained in an unsupervised fashion. In Section 4, we use a modified version of SfM-Net to pre-train the encoder of our MOREL technique.

## 3 Related Work

Previous works have used learned feature representations for reinforcement learning [7, 11, 10, 19]. Finn et al. [7] train a fixed feature extractor to encode key spatial feature points in images. They use this spatial encoding when training reinforcement learning algorithms on robotic manipulation tasks. Higgins et al. [11] use learned feature representations for sample efficient domain adaptation. By learning a disentangled representation of the environment, their reinforcement learning agent is able to quickly adapt to new domains.

### 3.1 Object-Based RL

Object representations are not new in reinforcement learning. Diuk et al. [5] proposed an object-oriented framework to facilitate reasoning about objects and their interactions in reinforcement learning. The framework was extended with a physics based model to reason about the dynamics of objects [29]. Other frameworks such as interaction networks [2] and schema networks [16] have been combined with reinforcement learning to reason and learn about causal relationships among objects. All of these frameworks can improve generalization and reduce sample complexity in reinforcement learning, but they assume that object features and relations are directly available from the environment. Li et al. [21] proposed object-sensitive deep RL by introducing a template matching technique to detect objects in raw images and augmenting them with object channels, but suitable templates must be handcrafted for each object in each environment. Bellemare et al. [3] used classic computer vision algorithms to detect objects in the Atari domain. Similarly, these hand-engineered methods rely on assumptions that do not hold outside of the Atari domain. In contrast, we propose an unsupervised technique that does not require any domain information, labeled data, or manual input from practitioners to automatically detect moving objects with their motion in visual reinforcement learning tasks.

### 3.2 Auxiliary objectives and model-based RL

Another approach to reduce the sample complexity of reinforcement learning consists of augmenting rewards with auxiliary objectives corresponding to auxiliary control or reward tasks [14, 22]. In navigation tasks, data efficiency can be improved with auxiliary depth prediction and loop closure classification tasks [23]. More generally, an agent capable of predicting future rewards and observations is extracting useful information that can speed up the learning of a good policy or value function [26]. This agent is also effectively learning a model of the environment based on which it can generate additional simulated data to learn without interacting with the environment [10]. However, there is a risk that the simulated data will harm the learning process if it is inaccurate. To that effect, Weber et al. [27] devised an approach to "interpret" simulated trajectories and try to extract only beneficial information. Our technique is orthogonal to this work and could be combined with auxiliary tasks and model-based techniques to further reduce sample complexity.

### 3.3 Interpretable RL

Khan et al. [17] pioneered a generic technique to explain policies for factored Markov decision processes in terms of the occupancy frequency of future states. This approach assumes discrete

features and a model of the environment. In deep learning, classifications and predictions can often be visualized by finding the input that maximizes some desired output [32]. Along those lines, a saliency map of the input features can also be used to explain the degree of sensitivity of each feature to a desired output. The saliency of each feature corresponds to the magnitude of the derivative of the output to that feature. Since such saliency maps may be difficult to interpret when the saliency of individual pixels do not produce recognizable images, Iyer et al. [13, 21] proposed object-level saliency maps to explain RL policies in visual domains by measuring the impact on action choices when we replace an object in the input image by the surrounding background. However, templates for each object must be hand-engineered. We show how the object masks produced by our technique in an unsupervised fashion can help to interpret and visualize the information used by RL policies.

## 4 Method

We decouple our training procedure into two phases. First, we learn a structured representation of the input that captures information about all moving objects in the scene through training on the task of unsupervised video object segmentation. After learning a disentangled representation of the scene, we transfer the weights to our reinforcement learning agent and continue to optimize our segmentation network jointly along with the policy and/or value function. This biases the model to focus on moving objects, allowing us to obtain a good policy, and requires fewer interactions with the environment in the second phase.

### 4.1 Unsupervised Video Object Segmentation

**Model Architecture**   Our architecture is illustrated in Figure 1. Given two consecutive frames $x_0, x_1 \in \mathbb{R}^{W \times H}$, our network predicts $K$ object segmentation masks $M^{(k)} \in [0, 1]^{W \times H}$, $K$ corresponding object translations $t_k \in \mathbb{R}^2$, and a camera translation $c \in \mathbb{R}^2$. We found $K = 20$ to work well. Our model compresses the input images to a 512-dimensional embedding which is forced to contain information about all of the moving objects in the input. We use the standard architecture from [24, 31, 34] which we modify to take in 2 frames as input instead of 4 to match the inputs of SfM-Net [35]. We upsample the embedding via a fully-connected layer followed by two more layers of bilinear interpolation and convolutions to predict the $K$ object masks. A sigmoid non-linearity ensures $M^{(k)} \in [0, 1]^{W \times H}$ while allowing each pixel to belong to many objects or no objects (indicating background). A separate branch from the embedding, consisting of a single fully-connected layer, computes the camera translation. Contrary to many segmentation networks [1, 28, 35], our model does not make use of skip connections from the downsampling path to the upsampling path, and this ensures that all object information will be present in the embedding.

**Reconstruction Loss**   Since we do not have ground truth for the object masks or translations, we follow the approach in [35] and train the network to interpolate between $x_0$ and $x_1$. Using the object masks and translations, we estimate the optical flow $F \in \mathbb{R}^{W \times H \times 2}$. This is computed as shown in Equation 1 by performing the pixel-wise sum of all the object translations weighted by the probability of that pixel belonging to the corresponding object. The camera translation $c$ is also added afterwards.

$$F_{ij} = \sum_{k=1}^{K} (M_{ij}^{(k)} \times t_k) + c \tag{1}$$

We use the optical flow to warp $x_1$ into an estimate of $x_0$, $\hat{x}_0$. This can be done differentiably using the method proposed by Spatial Transformer Networks [15]. We train the network to minimize this reconstruction error, as achieving a low reconstruction error requires the network to accurately model all the moving objects in the scene, allowing it to disentangle many factors of variation.

We use structural dissimilarity (DSSIM) [36], a perceptually-based distance metric for images, to train our network to reconstruct $x_0$. This loss is chosen instead of the L1 loss used to train SfM-Net [35]. The L1 loss suffers from a gradient locality problem [4] where each pixel's gradient is function of only its immediate neighbouring pixels [39]. This makes it difficult for the network to learn translations which are further away or that are in regions with little texture. DSSIM, on the other hand, is computed with an $11 \times 11$ filter that helps ensure that the gradient at each pixel gets signal

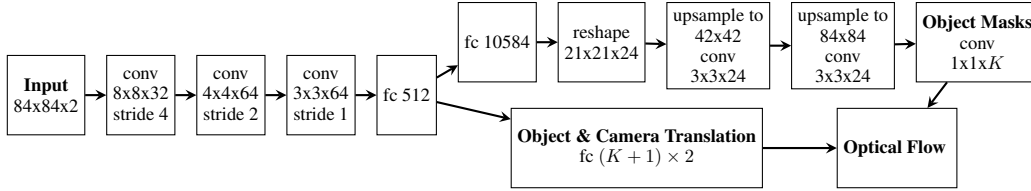

Figure 1: Unsupervised object segmentation model architecture. There are 3 convolutional layers which are followed by a fully-connected layer to compress the input into a 512-dimensional embedding. We use the embedding to predict object and camera translations. To predict object masks from the embedding, we have another fully-connected layer and then reshape the activations into a 3D volume. Next, we use bilinear interpolation to increase the size of the activations before applying a stride 1 convolution. After upsampling in this manner twice, the activation volume is the desired dimensionality for the object masks. Every convolution is followed by an ReLU non-linearity, except for the object masks which are followed by a sigmoid. The optical flow is computed using the translations and object masks as shown in Equation 1.

from a large number of pixels in its vicinity. In turn, this makes the optimization procedure more stable.

**Flow Regularization**    Solely minimizing reconstruction loss is insufficient to get sharp, interpretable object masks because the problem is underdetermined. The network can learn to predict many translations which cancel out to the correct optical flow. One approach to solve this is imposing L1 regularization on the object masks to encourage sparsity. However, if we only regularize the object masks, there is still a degenerate solution consisting of object masks with very small values coupled with large object translations. When multiplied, the computed optical flow is equivalent, however the L1 norm of the object masks will be very low. To get sparse object masks and reasonable translations, we impose L1 regularization only after multiplying each mask by its corresponding translation.

$$\mathcal{L}_{reg} = \sum_{k=1}^{K} \|M^{(k)} \times t_k\|_1 \tag{2}$$

$$\mathcal{L}_{seg} = \mathcal{L}_{reconstruct} + \lambda_{reg}\mathcal{L}_{reg} \tag{3}$$

**Curriculum**    We minimize $\mathcal{L}_{seg}$ where $\lambda_{reg}$ is a hyperparameter which determines the strength of the regularization. Setting $\lambda_{reg}$ too high causes the network to collapse and predict zero optical flow as this minimizes $\mathcal{L}_{reg}$. To resolve this training instability, for the first 100k steps we linearly increase $\lambda_{reg}$ from 0 to 1. This progressively makes the task harder and slowly encourages the network to make the object masks interpretable without collapsing.

**Training Procedure**    We collect 100k frames by following a random policy. Using an Adam optimizer [18] with learning rate $1 \times 10^{-4}$ and batch size 16, we minimize $\mathcal{L}_{total}$ for 250k steps.

### 4.2    Transferring for Deep Reinforcement Learning

**Model Architecture**    We present a motion-oriented actor critic algorithm. Our architecture is illustrated in Figure 2. To learn a good policy, our RL agent must receive information about both moving objects and static objects. Although our segmentation network provides a disentangled representation of the moving objects in a frame, it is not designed to capture static objects — which in some games can nonetheless be important for achieving high reward. To capture information about static objects, we add a downsampling network with the same architecture used in Figure 1. We concatenate the embeddings from the segmentation and the static object networks, and then add a fully-connected layer to combine this information and produce further higher-level features. From this, a final fully-connected layer outputs our policy and state value as in [24].

To initialize our RL agent, we transfer the weights in the motion path from our segmentation network and randomly initialize the weights of the static path. By only imposing a motion-oriented prior on one subnetwork, we allow the static network to learn complementary features.

**Joint Optimization** Our segmentation network is trained on a dataset which is collected by following a uniform random policy. It provides a strong prior with which to initialize the motion-oriented path of our reinforcement learning agent. During training, however, it is beneficial to continue updating the transferred weights on the object segmentation task. The benefits are threefold. Firstly, this allows the object segmentation path to continue improving as it sees more data from the environment. Secondly, it allows the agent to retain its ability to segment out objects, which can be useful for visualization. Finally, for many games, there is a distribution shift in the input that occurs as the agent's policy improves. Concretely, if a game has multiple levels, an agent that is continually improving will encounter levels that were difficult to reach via the random policy. On these inputs, the transferred weights from the segmentation network become less meaningful since these examples can be drastically different from the training set. Thus, we utilize joint optimization to update the agent by minimizing $\mathcal{L}_{seg}$ alongside the policy and value function in an online fashion.

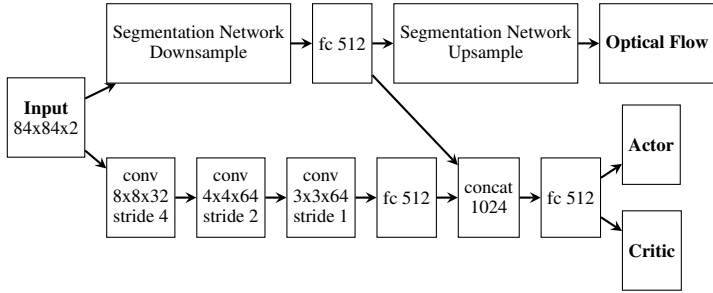

Figure 2: MOREL model architecture. We concatenate the results of our segmentation network (top) and static object network (bottom), and use this representation to predict a policy and state value function. The static object network uses the same network architecture as our segmentation network. The segmentation network downsampling and upsamping architecture is shown in Figure 1.

## 5 Experimental Results

We perform a qualitative evaluation of our unsupervised video object segmentation network to analyze the quality of the produced object masks. Then, we perform an ablation study to provide insight into our reinforcement learning results on a standard set of Atari games, as chosen in [24, 25, 38]. Finally, we show the results of MOREL on all 59 Atari games which were available at the time of this publication in our supplementary materials. Since our approach is orthogonal to the choice of RL algorithm, it can in principle be combined with any RL algorithm. To verify this empirically, we show results using two different reinforcement learning algorithms, A2C and PPO.

### 5.1 Unsupervised Video Object Segmentation

Figure 3 shows object masks and optical flow predicted by our unsupervised video object segmentation network. Since there are $K = 20$ object masks, we only display the most salient object mask individually, which we define as the one with the highest $\|M^{(k)} \times t_k\|_1$. This is the object mask with the highest flow regularization penalty, and consequently the one in which the model was most confident. To get an overall visualization of the 20 masks, we also display a weighted sum of the masks, where each mask is weighted by the L1-norm of its corresponding predicted object translation. This weighting corresponds to model confidence - since the flow regularization penalizes the network heavily when it predicts large translations, the model has more confidence that an object is there if the predicted translation is large.

Overall, our network picks out all the moving objects in the scene, a remarkable feat considering that our method is completely unsupervised. There are a few interesting observations about the object masks that are of note. Firstly, since many objects in Atari games have no texture, small translations

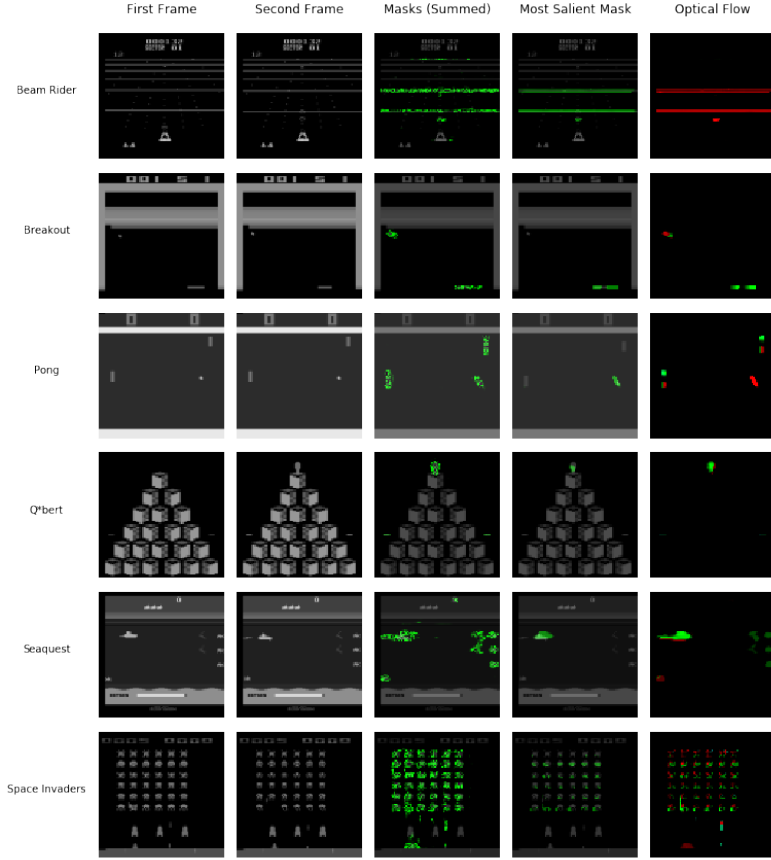

Figure 3: Unsupervised video object segmentation results. The first and second frames are the inputs to the network, pre-processed as in [24]. The masks are shown overlaid on the frame in green. The intensity of the green indicates the confidence of the model that there is an object present at each pixel. The hue of the optical flow image indicates direction and the intensity of the color denotes the magnitude of the flow (normalized per image).

of these objects do not actually change the pixels located at their centers. Correspondingly, the model is free to ignore these stationary portions of the object. In our Breakout results, the network learns to split the paddle into a left and right side, and does not move the middle portion. Another interesting quirk in Atari games is that separate objects are often programmed to move in the same way. In Space Invaders, many enemies traverse the screen in a single formation, and as a result our model merges them all on one object mask and treats them as a single entity. Finally, there are some Atari games where our model trains well but our fundamental assumption that motion is a helpful cue for understanding the game is not satisfied. On Beam Rider, our network achieves low loss after learning to segment the game's light beams, however these beams are purely visual effects and are unimportant for action selection. Consequently, the game enemies, which are much smaller in size, are ignored by the network, and the resulting learned representation is not that useful for a reinforcement learning agent. Further visualizations in the form of videos are available for all 59 Atari games in the supplementary material. The segmentations in each video provide insights into what the agent understands and whether it is making decisions based on sensible information.

## 5.2 Reinforcement Learning Experimental Setup

Following the experimental setup from [24], we train each agent for 10 million timesteps with one timestep for each frame. The hyperparameters for our A2C and PPO agents were taken from [24] and [31] respectively. PPO defines a surrogate loss function which uses clipping to control the size of policy updates [31]. When composing MOREL with PPO, we must also ensure that we limit the

magnitude of our weight updates after the addition of our segmentation loss during joint training. We achieve this by clipping our segmentation loss whenever the policy or value loss is clipped by PPO. Other than this change, our method is applied identically to A2C and PPO in our experiments.

## 5.3 Ablation Study

To provide insight into our algorithm, we ablate MOREL and compare against suitable baselines using A2C. The results are shown in Figure 4. We train three baselines on A2C. One baseline uses the standard architecture from [24] with randomly initialized weights. The second baseline uses the same architecture as our model, again initialized randomly, to control for any differences in architecture and the number of parameters. Finally, we compare to an agent initialized with weights transferred from an autoencoder [12] trained on the same dataset as our segmentation network. The autoencoder architecture is the same as our network, only modified to output one frame instead of $K$ object masks, and without the object translation and camera motion prediction. We also follow the same training procedure while minimizing the L2 reconstruction error instead.

We ablate MOREL in two steps while comparing to the baselines. First, we do a simple transfer of the object segmentation network, and use only the motion path without joint training. Then, in addition to transferring the weights, we add in joint optimization using the architecture from Figure 2.

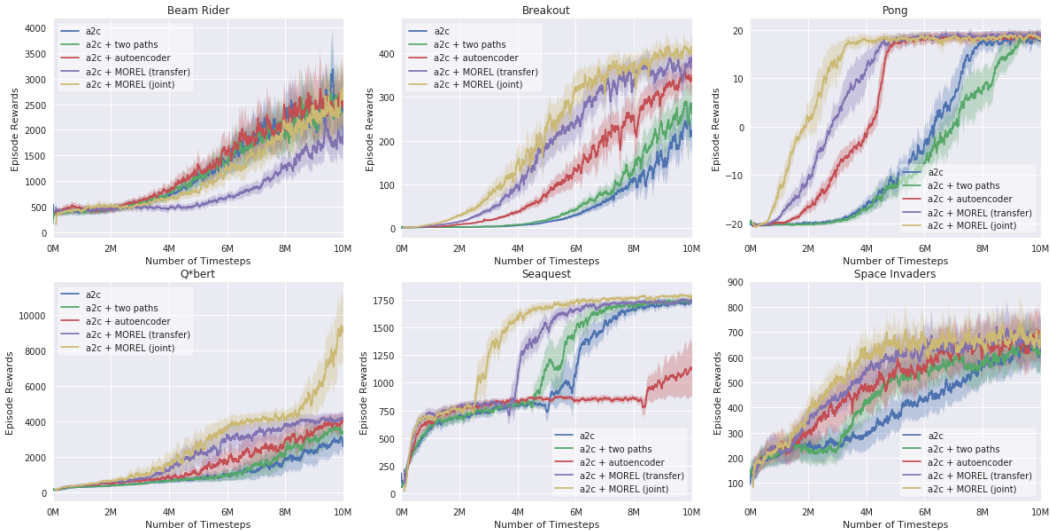

Figure 4: Comparison of our approach against several baselines over 3 random seeds. Methods which were pre-trained are shown offset by 100k frames to account for the size of the pre-training dataset.

Overall, we see empirically that MOREL significantly improves the sample complexity of A2C. Without our object segmentation method, the architecture shown in Figure 2 does not perform better than the baseline, despite having twice the number of parameters. Additionally, we see that our approach provides a superior prior compared to a trained autoencoder. We did not expect MOREL to perform well on Beam Rider because our assumption that moving objects are relevant to the policy is not satisfied. We hypothesize that a different learned prior would be needed to improve sample complexity. However, it is encouraging that even then, MOREL with joint optimization performs no worse than the baseline. MOREL's strong performance on Q*bert also provides a strong case for joint training. On Q*bert, the second level is never reached in the pre-training dataset when executing random actions. However, the agent eventually reaches this level, and here joint optimization offers a strong performance boost by correcting for the shift in input distribution caused by this new level.

Figure 5 shows the results of MOREL where each modification that we did to SfM-Net is removed individually. This demonstrates the necessity of the curriculum, flow regularization, and using DSSIM as a loss function.

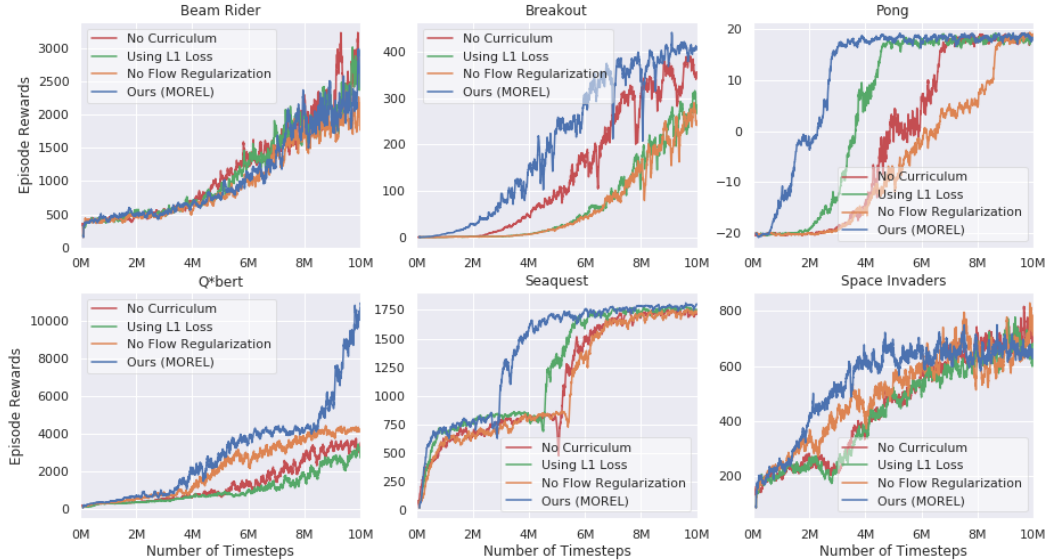

Figure 5: Comparison of our approach where MOREL is modified to remove each improvement that we did to SfM-Net.

## 5.4 Broad comparison

In the supplementary material, we showcase the performance of MOREL on all 59 Atari games when combined with A2C (Figures 5 and 6) and PPO (Figures 7 and 8). We observe a notable improvement on the A2C runs for 26 of the games, and a worse performance in only 3 games. For PPO, composition with MOREL provides a benefit on 25 games, and a deterioration in 9 games.

## 6 Conclusion

In summary, this paper describes an unsupervised technique for object segmentation and motion estimation in reinforcement learning from raw images. This approach reduces the amount of interaction with the environment when moving objects are important features. The accuracy of the resulting object masks also provide support to interpret and explain the performance of some policies. In future work, we plan to extend this framework to fixed objects and explicitly learn about salient (fixed or moving) objects with an attention model. It would also be interesting to combine this work with object-oriented frameworks [21], physics-based dynamics [29] and model-based reinforcement learning [27] to reason about and plan object interactions. We plan to further extend this work to 3D environments such as Doom and real-world environments (e.g., autonomous vehicles and robotics).

#### Acknowledgments

We would like to thank Francois Chaubard for his helpful discussions about loss functions and joint optimization techniques. Also, this work benefited from the use of the CrySP RIPPLE Facility at the University of Waterloo.

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
