[Supplementary Material]

# 7   Performance on More Atari Games

In our supplementary work, we include one video for each Atari game showing our final trained MOREL agent with PPO. The videos are available at `https://www.youtube.com/playlist?list=PLhh84l5EzbN4kPAkieJ6Il2l8EC4iw12W`. The game frames are shown on the left. On the right, we have overlaid the predicted object masks onto the frames using the same processing as in Figure 3. Watching our agent's actions along with the segmented objects can offer insight into what the agent is able to understand and how it makes its decisions. As opposed to a black-box neural network which cannot offer the same visibility into an agent's chosen actions, MOREL provides an added benefit of interpretability.

Additionally, we showcase the performance of MOREL on all 59 Atari games when combined with A2C (Figure 6 and Figure 7) and PPO (Figure 8 and Figure 9). We observe a notable improvement on the A2C runs for 26 of the games, and a worse performance in only 3 games. For PPO, composition with MOREL provides a benefit in 25 games, and a decrease in performance in 9 games.

Figure 6: Comparison of our joint optimization approach to using A2C in all of the available Atari environments (first half). MOREL is shown at an offset of 100k frames to account for the size of its pre-training dataset.

Figure 7: Comparison of our joint optimization approach to using A2C in all of the available Atari environments (second half). MOREL is shown at an offset of 100k frames to account for the size of its pre-training dataset.

Figure 8: Comparison of our joint optimization approach to using PPO in all of the available Atari environments (first half). MOREL is shown at an offset of 100k frames to account for the size of its pre-training dataset.

Figure 9: Comparison of our joint optimization approach to using PPO in all of the available Atari environments (second half). MOREL is shown at an offset of 100k frames to account for the size of its pre-training dataset.