[Reviews · NeurIPS 2018]

Reviewer 1



************************* Summary: ************************* This paper describes MOREL, an approach for automatically segmenting objects for reinforcement learning. In particular, this work uses SfM-Net [1], which learns to predict optical flow of a single image, to segment the objects in a state, and then uses this object-mask for reinforcement learning. MOREL is evaluated on all 59 Atari games, where it outperforms the baselines in several environments. ************************* Quality: ************************* This approach appears to be sound. Additionally, it is evaluated within several environments. The paper does not have enough discussion about weaknesses of the approach. Additionally, while there is a clear advantage of using MOREL over the baseline approaches, it would be better to have more than three averaged results. ************************* Clarity: ************************* This paper is well-written. I did not have any problems understanding the approach or motivation. I did not even find any typos. I do have a couple of comments, though. In the background, the policy is defined w.r.t. histories, but this is not a common way to formulate policies. It is unclear why it is defined in this way in the paper. Additionally, r is never defined. ************************* Originality: ************************* The main contribution of this work is using SfM-Net to segment images for reinforcement learning. This is similar to feature extraction, and other similar approaches should have been discussed or compared to. In general, the related work lacks discussion of related feature construction and computer vision techniques. Atari is simple and the features could be constructed using very basic computer vision techniques. In fact, Bellemare et al. [2] list a many ways to extract features for Atari, for example by using background subtraction. Additionally, DARLA [3] uses Beta-VAE to learn disentangled features, such as objects, for reinforcement learning, and deep spatial auto-encoders [4] also learn features that encode information about objects. ************************* Significance: ************************* This work introduces an interesting way to segment objects for reinforcement learning. Vijayanarasimhan et al [1] demonstrate that Sfm-net can work on realistic environments, but it would still be useful to also evaluate MOREL within less trivial environments, particularly because the backgrounds in Atari games are stationary. This would make it clear that it is beneficial to segment objects. For example, do the masks even need to be learned? Can the objects be detected by computing optical flow of the previous and current frames or background subtraction? Can pre-trained features be used? Additionally, the paper described few changes to SfM-Net. It would have been nice to see results for the unchanged network. The main changes were a different loss for reconstruction, flow regularization, and a curriculum. These changes are incremental, and so the segmentation results from Figure 3 seem more like result from running SfM-Net on Atari, rather than any contribution from this work. This statement could be argued against by demonstrating that the adjustments to SfM-Net yield different or better results than the unchanged network. Nevertheless, the approach performs well against the baselines, and I do believe it could be a useful approach since it would likely make it easier to learn about salient objects. ************************* Comments ************************* The results do not demonstrate that this representation is more interpretable. Detecting objects does not give any insight into the decisions being made by the policy. Using attention might change this. Line 234: I disagree that the segmentation is remarkable since the approach is only evaluated on Atari. Additionally, the model is self-supervised. It is given optical flow and thus predicts optical flow. So the result does not seem too surprising. Why should we assume that moving objects are important? ************************* Typos ************************* None, well done! ************************* References: ************************* [1] Vijayanarasimhan, Sudheendra, et al. "Sfm-net: Learning of structure and motion from video." arXiv preprint arXiv:1704.07804 (2017). [2] Bellemare, Marc G., et al. "The arcade learning environment: An evaluation platform for general agents." Journal of Artificial Intelligence Research 47 (2013): 253-279. [3] Higgins, Irina, et al. "Darla: Improving zero-shot transfer in reinforcement learning." arXiv preprint arXiv:1707.08475 (2017). [4] Finn, Chelsea, et al. "Deep spatial autoencoders for visuomotor learning." arXiv preprint arXiv:1509.06113 (2015). Update: I have read the review's and author's response. While I appreciate the rigorous studies done in the atari environment, I still believe the paper would be greatly improved if it demonstrated the benefits of the approach in environments with more difficult segmentation. This could still be done in simulation. As such, my score remains the same.

Reviewer 2



This paper proposes to incorporate unsupervised object segmentation networks into deep reinforcement learning. The segmentation network is based on SfM-Net and can be jointly trained with the policy. I think this is an interesting idea and seems to be working well on standard setups. Because of this, I recommend acceptance for this paper. My main concern is the comparison to attention-based RL. The idea of using object mask seems to be very similar to using attention, which is not discussed enough and compared in the experiment. The main difference is that the object mask in this paper uses the strong prior that "objects" are useful, while attention can be fully data-driven. I'm wondering how does attention-based RL compare to this work, and particularly with some form of object mask pretraining/fine-tuning.

Reviewer 3



This paper proposes a novel moving object oriented deep RL method. The new method first performs unsupervised segmentation to the moving objects in a video, and then learns better policy and value based on the obtained moving object segmentation mask and optical flow. During the actor-critic model learning, the segmentation network is also further updated to better segment the moving objects when the training data changes in distribution. Pros: 1. The motivation of object oriented RL is reasonable and should be promising. 2. The proposed method is based on unsupervised learning which utilizes flow information and reconstruction constraint, and does not require labeled data. This is useful in the practical usage. 3. Experimental results show the superiority of the method over previous ones. Cons: The experiment section is somewhat weak. First, the method is only evaluated on Atari game. More virtual gyms can be used to run the proposed method but not been tried. Second, more ablation studies are desired, e.g. the effects of flow regularization, the use of curriculum by linearly increasing reg coefficients.